# Molecular Cloning and Expression Analysis of Three Suppressors of Cytokine Signaling Genes (*SOCS5*, *SOCS6*, *SOCS7*) in the Mealworm Beetle *Tenebrio molitor*

**DOI:** 10.3390/insects10030076

**Published:** 2019-03-16

**Authors:** Bharat Bhusan Patnaik, Bo Bae Kim, Yong Hun Jo, In Seok Bang

**Affiliations:** 1Division of Plant Biotechnology, Institute of Environmentally-Friendly Agriculture (IEFA), College of Agriculture and Life Sciences, Chonnam National University, Gwangju 61186, Korea; drbharatbhusan4@gmail.com (B.B.P.); kbb941013@gmail.com (B.B.K.); 2School of Biotech Sciences, Trident Academy of Creative Technology (TACT), Chandrasekharpur, Bhubaneswar, Odisha 751024, India; 3Department of Biological Science and the Research Institute for Basic Sciences, Hoseo University, Asan 31499, Korea

**Keywords:** *Tenebrio molitor*, suppressor of cytokine signaling, insect immunity, gene expression

## Abstract

Suppressors of cytokine signaling (SOCS) influence cytokine and growth factor signaling by negatively regulating the Janus kinase (JAK)-signal transducers and activators of transcription (STAT) pathway to maintain homeostasis during immune responses. However, functional characterization of SOCS family members in invertebrates is limited. Here, we identified and evaluated three SOCS genes (type I sub-family) in the mealworm beetle *Tenebrio molitor*. The full-length open reading frames (ORFs) of TmSOCS5, TmSOCS6, and TmSOCS7 comprised of 1389, 897, and 1458 nucleotides, encoding polypeptides of 462, 297, and 485 amino acids, respectively. The SH2 and SOCS box domains of the TmSOCS C-terminal region were highly conserved. Phylogenetic analysis revealed that these SOCS genes were clustered within the type I subfamily that exhibits the highest amino acid identity with *Tribolium castaneum* SOCS genes. Contrary to TmSOCS7 expression, the expression levels of TmSOCS5 and TmSOCS6 were lower in the larval, pupal, and adult stages. In larvae and adults, the expression levels of TmSOCS5 and TmSOCS6 were highest in the hemocytes and ovaries, respectively. SOCS transcripts were also highly upregulated in the hemocytes of *T. molitor* larvae within 3–6 h post-infection with the fungus *Candida albicans*. Collectively, these results provide valuable information regarding the involvement of TmSOCS type-I subfamily in the host immune response of insects.

## 1. Introduction

Cytokines are secretory proteins that regulate inflammatory responses. Most cytokines promote gene expression through the Janus kinase (JAK)-signal transducers and activators of transcription (STAT) pathway. Cytokine signaling and the JAK-STAT pathway are known to play essential roles in metazoan development and homeostasis of immune responses [1,2]. A large number of JAKs and STATs (e.g., four JAKs and seven STATs in humans) exist, and there appears to be differential employment of specific JAK-STAT pathway components in response to signaling mediated by a plethora of cytokine molecules. Non-redundant JAK/STAT signaling occurs in mammals, with preferential usage of various JAKs and STATs in response to specific cytokines/growth factors. Additionally, multiple regulators tightly control cytokine-mediated cellular effects by limiting inappropriate activity and the development of related diseases. Various proteins, including suppressor of cytokine signaling (SOCS) family members (a specific SH2-domain-containing tyrosine phosphatase [SHP] family); and protein inhibitors of STATs (PIAS), function as negative regulators of the JAK-STAT pathway, and help maintain homeostasis during defense reactions [3,4,5].

SOCS family proteins are prime regulators of the JAK-STAT pathway. They control inflammatory signaling by acting as pseudo-JAK substrates, blocking STAT signaling, and directing multiple pathway components towards ubiquitin-mediated proteasomal degradation [6]. Four members of the SOCS family, including SOCS1, SOCS2, SOCS3, and cytokine-inducible SH2-containing protein (CISH), were originally identified in mice [7]. These proteins negatively regulate cytokine induction, by either competing with STAT for binding to the cytoplasmic domains of the phosphorylated receptors or by inactivating the enzymatic activity of JAKs [8]. Subsequently, four additional members (SOCS4, SOCS5, SOCS6, and SOCS7) were identified in mice based on their conserved central SH2 and C-terminal SOCS-box domains. Evolutionary analysis of SOCS family proteins revealed a clear division, with CISH, SOCS1, SOCS2, and SOCS3 grouped to the type II subfamily, and the rest of the members grouped within the type I subfamily [9]. SOCS4 and SOCS5 are close homologs that are involved in a negative feedback loop for epidermal growth factor receptor (EGFR) signaling by directing the degradation of EGFR in a ligand- and E3 ubiquitin ligase c-Cbl-independent manner in cultured CHO cells [4]. SOCS6 associates with and inhibits the insulin receptor and is related to cytokine-mediated insulin resistance in SOCS6-overexpressing transgenic mice, while human SOCS7 interacts with STAT5 or STAT3, to prevent nuclear translocation and to attenuate prolactin, growth hormone, and leptin signaling [10,11]. Eight members of the SOCS family have been identified in vertebrates; however, the SOCS gene repertoire is greatly expanded in rainbow trout, with 26 expressed genes within the type I and type II subfamilies. This has been attributed to the expansion of type II SOCS genes from a single CISH/SOCS1-3 precursor via whole genome duplication events. [12].

Knowledge of SOCS gene families in invertebrates is limited to a few isolated reports from molluscan and arthropod species. In *Drosophila*, SOCS36E, which shares 64% identity with human SOCS5, was the first SOCS protein identified, and its expression during embryogenesis was studied [13]. Later, it was shown to function as a negative regulator of the JAK/STAT and EGFR pathways. In contrast, two identified genes, SOCS44A (34% and 33% identity with human SOCS6 and SOCS7, respectively) and SOCS16D (48% and 45% identity with human SOCS6 and SOCS7, respectively), exhibited limited involvement in the JAK/STAT cascade, although SOCS44A was considered to be a transcriptional target of STAT92E [14]. Additionally, in the pacific oyster (*Crassostrea gigas*), three SOCS genes (SOCS2, SOCS5, and SOCS7) have been identified as putative inducers of NF-κB transcription. SOCS2 homologues exhibiting immune-related expression have also been identified in *Ruditapes philippinarum* [15], *Haliotis discus* [16], *Eriocheir sinensis* [17], *Procambarus clarkii* [18], and *Litopenaeus vannamei* [19]. Further, direct evidence suggests that SOCS6 plays a role in activation of the NF-κB signaling pathway in *E. sinensis* [20]. The *Bombyx mori* SOCS2 homolog has been demonstrated to function as a negative regulator of the JAK/STAT and ecdysteroid signaling pathways [21], while the SOCS6 homolog regulates the EGFR pathway [22]. The type I SOCS genes, however, have not been previously reported in coleopteran insects. In the present study, we identified three type I SOCS genes (SOCS5, SOCS6, and SOCS7) in the coleopteran model insect *Tenebrio molitor* through bioinformatics analysis. We also cloned the SOCS homologs and characterized their evolutionary relationship through phylogenetic analysis. Finally, we studied their expression profiles in response to exposure to microorganisms, which provided insights into the putative immune functions of SOCS in insects.

## 2. Materials and Methods

### 2.1. Insect Rearing

*Tenebrio molitor* were reared in the laboratory on an artificial diet (1.1 g of sorbic acid, 1.1 mL of propionic acid, 20 g of bean powder, 10 g of brewer’s yeast powder and 200 g of wheat bran in 4400 mL of DW; autoclaved at 121 °C for 15 min) in the dark at 26 ± 1 °C, under 60% ± 5% relative humidity.

### 2.2. Microorganisms

The Gram-negative bacterium *Escherichia coli* (strain K12), Gram-positive bacterium *Staphylococcus aureus* (strain RN4220), and the fungus *Candida albicans* were used for the immune challenge experiments. *E. coli* and *S. aureus* were cultured overnight in Luria Bertani (LB) broth at 37 °C, and *C. albicans* was cultured in Sabouraud Dextrose broth. The microorganisms were harvested, washed twice in phosphate-buffered saline (PBS; pH 7.0), and centrifuged at 3500 rpm for 10 min. The samples were then suspended in PBS, and the optical density was measured at 600 nm (OD_600_). *E. coli* and *S. aureus* were diluted to 10^6^ cells/μL, and *C. albicans* was diluted to 5 × 10^4^ cells/μL for the immune challenge studies.

### 2.3. Identification and In Silico Characterization of TmSOCS Genes

The sequences of TmSOCS5, TmSOCS6, and TmSOCS7 were retrieved from the *T. molitor* RNAseq (unpublished) and expressed sequence tag (EST) databases by using local-tblastn analysis [23]. For the retrieval, the amino acid sequences of *T. castaneum* SOCS5 (XP_015833441.1), SOCS6 (XP_008190646.1), and SOCS7 (XP_008190646.1) were used as queries. Conserved domains were identified using InterProScan (http://www.ebi.ac.uk/interpro/search/sequence-search) and blastx (https://blast.ncbi.nlm.nih.gov/Blast.cgi). A domain-specific multiple alignment that included representative SOCSs from other insects retrieved from GenBank was generated using Clustal X2 [24]. The percent identity and phylogenetic analyses were performed using Clustal X2 and MEGA7 [25], respectively. The evolutionary relationships were inferred by the neighbor-joining method [26]. The bootstrap consensus tree was inferred from 1000 replicates, and the evolutionary distances were computed using the Poisson correction method. Human SOCS1, which belongs to the type II subfamily, was used as an outgroup.

### 2.4. Cloning the TmSOCS ORF

Primers were designed to amplify the TmSOCS ORFs based on the identified sequences, and the ORFs were amplified by polymerase chain reaction (PCR) using AccuPower^®^ PyroHotStart Taq PCR PreMix (Bioneer, Korea). The primers are listed in Table 1. Briefly, 1 µg of RNA was used as the template to synthesize cDNA using Oligo(dT) primers. Then, the generated cDNA was diluted 20 times, and 1 µL was used for PCR amplification under conditions that included an initial denaturation step at 95 °C for 5 min, followed by 35 cycles of 95 °C for 30 s, 53 °C for 30 s, and 72 °C for 2 min. The PCR products were purified by the AccuPrep^®^ PCR Purification Kit (Bioneer, Korea) and immediately ligated into a T-Blunt vector (Solgent, Korea). The PCR product-vector ligation was transformed into *E. coli* DH5α competent cells according to the manufacturer’s instructions. After validation by colony PCR, the plasmid DNA was extracted from the cells using the AccuPrep^®^ Nano-Plus Plasmid Extraction Kit (Bioneer, Korea), and the cloned ORF was sequenced by using the M13 forward and reverse primers (Table 1).

### 2.5. Developmental and Tissue-Specific Expression of TmSOCS Transcripts

Total RNA was isolated from eggs, early larvae (12th–15th instar; length, ~2.4 cm), late larvae (19th–20th instar; length ~3 cm), pre-pupae, 1–7-day-old pupae, and 1–5-day-old adult insects to monitor the expression of TmSOCS5, TmSOCS6, and TmSOCS7 during development. The pupal (PP to P7) and the adult stages (A1 to A5) of *T. molitor* considered for development expression were not sexed before sampling. For the tissue-specific expression analysis, total RNA was isolated from the hemocytes, gut, Malpighian tubules, fat body and integument of late-instar larvae. Total RNA was also isolated from the ovaries and testes of 5-day-old adults. The LogSpin RNA isolation method [27], with minor modifications, was used to isolate total RNA from the tissue and whole-body samples. Briefly, the samples were homogenized in 1 mL of guanidine thiocyanate RNA lysis buffer (20 mM EDTA, 20 mM MES buffer, 3 M guanidine thiocyanate, 200 mM sodium chloride, 0.005% Tween-80, and 1% isoamyl alcohol in DW, pH 5.5), and centrifuged at 21,000× *g* for 5 min at 4 °C. After a 1 min incubation in absolute ethanol, the samples were transferred to silica spin columns, and centrifuged at 21,000× *g* for 30 s at 4 °C to remove debris. After DNase treatment and two washes with 3 M sodium acetate and 80% ethanol, the total RNA was eluted with DNase and RNase-free water. The cDNAs were synthesized from 2 μg of total RNA using AccuPower^®^ RT PreMix (Bioneer, Korea) and Oligo(dT)_12–18_ primers on a MyGenie96 Thermal Block (Bioneer, Korea). Then, the developmental and tissue-specific expression of the TmSOCS genes was analyzed by quantitative real-time reverse transcriptase polymerase chain reaction (qRT-PCR). The qRT-PCR assay was performed in a 20-µL reaction containing AccuPower^®^ 2X GreenStar qPCR Master Mix (Bioneer) and primers for TmSOCS5, TmSOCS6, and TmSOCS7 (Table 1). The cycling parameters included an initial denaturation step at 95 °C for 5 min, followed by 40 cycles of 95 °C for 30 s, 60 °C for 30 s and 72 °C for 40 s. The qRT-PCR assays were performed on an AriaMx real-time PCR system (Agilent Technologies, Santa Clara, CA, USA) and the results analyzed using AriaMx real-time PCR software. The *T. molitor* Ribosomal protein L27a gene (*TmRpL27a*) was used for normalization, and the results were calculated by the ΔΔCt method [28]. The reactions were performed in triplicate, and the results represent mean ± S.E. of three biological replications.

### 2.6. Immune Challenge Studies

Healthy *T. molitor* larvae were administered with a 1 µL suspension containing either 10^6^ cells of *E. coli* or *S. aureus* or 5 × 10^4^ cells of *C. albicans* by intra-abdominal injection. The preparation methods for microorganisms and pretreatment methods were performed according to one of our previous studies [29]. The dose of microorganisms in the healthy larvae has been validated to be sufficient for immune challenge studies and the mortality of the larvae was limited to 10% or less in most cases. A similar volume of phosphate buffered saline (PBS) was injected into a separate group of larvae as a wounded control. Due to their pivotal roles in the humoral and cell-mediated immune responses, hemocytes, fat body and gut tissues were collected at 3, 6, 12, and 24 h post-injection to analyze the expression of TmSOCS5, TmSOCS6, and TmSOCS7 after microorganism challenge. Total RNA isolation, cDNA synthesis, and qRT-PCR were performed as described above. To determine the effects of microbial challenge on the expression of TmSOCS transcripts, the fold-change at each time point was determined by comparison to that of the PBS-injected control. All data are presented as mean ± standard error (SE). One-way analysis of variance (ANOVA) and Tukey’s multiple range tests were used to evaluate the differences between groups (*p* < 0.05). All statistical tests were performed using the Statistical Analysis Software (SAS) suite (SAS Institute Inc., Cary, NC, USA).

## 3. Results and Discussion

### 3.1. Identification of TmSOCS Homologs and Molecular Characterization

A local tblastn search was conducted using TcSOCS5, TcSOCS6, and TcSOCS7 to retrieve SOCS gene family sequences from the *Tenebrio* RNAseq and EST libraries, and the retrieved gene sequences were named TmSOCS5, TmSOCS6, and TmSOCS7, respectively. The putative ORF sequences for the TmSOCS genes were identified using the gene-finding program FGENESH (http://www.softberry.com/berry.phtml?topic=fgenesh&group=programs&subgroup=gfind). Next, primers were designed to amplify and clone the ORF sequences of TmSOCS5, TmSOCS6, and TmSOCS7. The TmSOCS constructs (T-blunt vector + full-length TmSOCS ORF insert) were sequenced to validate the TmSOCS genes. The sequences of TmSOCS5, TmSOCS6, and TmSOCS7 (nucleotide and translated protein sequences) were submitted to GenBank (NCBI) under the accession numbers MK292064, MK292065, and MK292066, respectively. The genomic architecture (exons and introns) of the *T. molitor* SOCS family genes was difficult to determine due to lack of a fosmid DNA library.

The TmSOCS5 ORF is 1389 bp and encodes a polypeptide of 462 amino acids (Figure 1). The computed molecular mass and isoelectric point of TmSOCS5 protein are 52.8 kDa and 6.23, respectively. These values are similar to those of the SOCS5 homolog found in *C. gigas,* which has a molecular mass of 51.5 kDa and an isoelectric point of 6.40 [30]. TmSOCS5 also contains an SH2 domain (residues 301–379) that includes a phosphotyrosine hydrophobic binding pocket and a C-terminal SOCS box domain (residues 381–437) containing the putative elongin B/C interaction residues. The TmSOCS6 ORF is 897 bp in length and encodes a polypeptide of 297 amino acids(Figure 2). Its calculated molecular mass and isoelectric point is 34.5 kDa and 9.40, respectively. The full-length TmSOCS6 sequence is shorter in comparison to the SOCS6 homologue found in *E. chinensis* [20], although typical SH2 and SOCS box domains are present. The TmSOCS7 ORF is 1458 bp in length and encodes a polypeptide of 485 amino acid residues (Figure 3). Its predicted molecular mass is 56 kDa, and its theoretical isoelectric point is 9.41. Similar to other SOCS family members, the phosphotyrosine, hydrophobic residues, and putative elongin B/C interaction sites are found in the conserved SH2 and SOCS box domains. This study and other closely-related studies have shown that the sequences in the amino-terminal region are diverse, while the central SH2 and carboxyl-terminal SOCS box domains are well conserved [31,32]. The SOCS box amino acid consensus sequence has also been found in protein families containing WD-40 repeats (IPR017986), ankyrin repeats (IPR020683), and SPRY domains (IPR003877) [8]. The elongin B/C region within the SOCS box domain may be involved in the negative regulation of signaling pathways by directing the target proteins for ubiquitination and proteasomal degradation [33,34].

The multiple sequence alignment (MSA) and percent identity matrix of the C-terminal region of TmSOCS5 and included insect orthologs are shown in Figure 4. The full-length amino acid sequences of the *T. molitor* SOCS proteins were not included in the phylogenetic analyses due to high level of sequence divergence within the N-terminus. As indicated in previous studies, the N-terminal region of SOCS proteins is largely disordered, and a region of ~70 residues is conserved across different species and between SOCS4 and SOCS5 [35,36]. Therefore, alignment of the conserved SH2 and SOCS box domains was deemed appropriate for understanding the sequence diversity and evolutionary position of *T. molitor* SOCS type I subfamily proteins. The SH2 domain contains numerous identical residues across species, whereas the SOCS box domain possesses lower, but still high levels of similarity (Figure 4A). TmSOCS5 is most similar to TcSOCS5 (97% identity), followed by OtSOCS5 from the dung beetle, *Onthophagus taurus* (88% identity) (Figure 4B). TmSOCS6 exhibited less similarity in the SH2 and SOCS box domains, but it was most similar to TcSOCS6 (91% identity; Figure 5). This protein exhibited significantly less similarity to other insect orthologs, with percent identities of 35%–55%. BmSOCS6 however, exhibited sequence similarity to lepidopteran SOCS6 family members, and *E. sinensis* SOCS6 possessed high levels of sequence similarity to other molluscan SOCS family members [20,21]. In contrast, high levels of sequence identity and similarity were observed in the SH2 and SOCS box domains of the SOCS7 homologs (Figure 6A). Interestingly, the SH2 and SOCS box domains of TmSOCS7 possessed 100% identity to TcSOCS7, and 74%–86% identity to other insect species (Figure 6B). Additionally, TmSOCS5, TmSOCS6, and TmSOCS7 were closely related to TcSOCS5, TcSOCS6, and TcSOCS7, respectively.

### 3.2. Molecular Evolutionary Relationships of the TmSOCS Proteins

To explore the phylogenetic relationships of the *T. molitor* type I SOCS proteins, we constructed a phylogenetic tree using the amino acid sequences of insect SOCS proteins (Figure 7). Human SOCS1, which belongs to the type II subfamily, was used as an outgroup. The phylogram indicated three conspicuous clusters of type II SOCS proteins comprising of the SOCS5, SOCS6, and SOCS7 proteins. The SOCS5 and SOCS7 proteins formed more compact clusters than those of SOCS6 cluster. As expected, TmSOCS5, TmSOCS6, and TmSOCS7 were closely related to TcSOCS5, TcSOCS6, and TcSOCS7, respectively. Additionally, the SOCS5 protein from another coleopteran species, OtSOCS5 was found to be closely related to TmSOCS5 and TcSOCS5. The TmSOCS6 cluster consisted of two sub-clusters. We described earlier, through MSA and percent identity analysis, that the SH2 and SOCS box domains of TmSOCS6 proteins are comparatively less well conserved than those of the TmSOCS5 and TmSOCS7 proteins. Also, the longer N-terminal region of the insect SOCS6 proteins, similar to vertebrate SOCS6 proteins, is thought to be involved in nuclear localization of this protein [37]. These results are consistent with similar report examining *E. sinensis* SOCS6 [20]. The observed close relationship of *Bombyx mori* SOCS6 (BmSOCS6) and *Danaus plexippus* SOCS6 (DppSOCS6) proteins is also consistent and is consistent with previous findings [22].

### 3.3. Developmental Expresion of TmSOCS Genes

We used qRT-PCR analysis to quantify TmSOCS5, TmSOCS6, and TmSOCS7 mRNA expression across the developmental (egg, early larva, late larva, pre-pupa, pupa [days 1–7] and adult [days 1–5]) stages and within various tissues (Figure 8). We hypothesized that there may be shifts in SOCS gene expression during the transitions between developmental stages and within a single developmental stage. Genome-wide screens have earlier been used to analyze differential expression of genes during different stages of insect development to understand the mechanisms of genic interplay regulating insect development [38,39]. TmSOCS5, TmSOCS6, and TmSOCS7 gene sequences were also retrieved from the *T. molitor* larval transcriptome database. Here, we were interested to note the expression of TmSOCS during *T. molitor* developmental stages. Our results indicate that TmSOCS5 and TmSOCS6 expression levels were higher at the egg stage. Relative to the expression levels of TmSOCS5 and TmSOCS6 in the egg stage, the mRNA expression in the larval, pupal, and adult stages of the insect were found to be lower. (Figure 8A,B). Moreover, we found a more consistent expression of TmSOCS5 and TmSOCS6 at the metamorphosing stages. In contrast, TmSOCS7 expression levels were relatively higher at the larval, pupal, and adult stages in comparison to that of the egg stage (Figure 8C). Further, in the present study, the adult insects used for the expression study were not sexed, and therefore a sex-specific relationship of TmSOCS genes could not be established. Such a relationship has been previously observed in the mosquito (*Anopheles culicifacies*), where SOCS mRNA was found to be highly expressed in the male relative to expression in female adults. It was also observed that although biases in the expression of genes begin at the larval stage, the expression of SOCS genes does not change between male and female larvae [40]. The involvement of TmSOCS gene in sexual dimorphism is more emphasized in the tissue-specific expression analysis of *T. molitor* adults. *Drosophila* SOCS36E, which has high levels of sequence identity with vertebrate SOCS5 (75% and 44% in the SH2 and SOCS-box domains, respectively), is known to be expressed during embryogenesis (especially during embryonic and imaginal disc development) [13]. Additionally, SOCS36E regulates STAT activity levels through either Cullin2 (Cul2) scaffolding protein-dependent or independent mechanisms in the egg chamber of *Drosophila* [41]. Relative expression levels of the *A. culicifacies* SOCS gene (AcSOCS) were also high at the egg stage signifying JAK-STAT control of embryogenesis [40]. Further, we assumed that SOCS5 and SOCS6 could be an important player in the embryogenesis process, but these proteins would exert limited function in the innate immunity of *T. molitor*. Preferentially, SOCS7 could be more involved in cellular homeostasis mechanisms during infection and immunity processes in the host insect. Supporting evidence reflects the involvement of SOCS7, and not SOCS5, in triggering the activation of the NF-κB signalling pathway in the mollusc, *C. gigas* [30].

### 3.4. Tissue-Specific Expression of T. molitor SOCS Genes

The tissue-specific expression analysis of *T. molitor* larvae indicated that the mRNA levels of TmSOCS5 and TmSOCS6 were higher in the hemocytes than those observed in the other tissues (Figure 9AI,BI), while TmSOCS7 levels were elevated in Malpighian tubules (Figure 9CI). High expression levels in hemocytes were also observed for SOCS5 and SOCS7 mRNA in the pacific oyster *C. gigas* [30] and for SOCS6 mRNA in the Chinese mitten crab *E. sinensis* [20]. Hemocytes and fat bodies were the principal target tissues for the expression of SOCS6 mRNA in *B. mori* [22]. In adults, the expression levels of TmSOCS5, TmSOCS6, and TmSOCS7 transcripts were higher in the ovaries (Figure 9AII, BII, and CII). This indicates that *T. molitor* SOCS transcripts show sexually dimorphic gene expression patterns. It is interesting to observe the active involvement of TmSOCS in the ovarian processes. We understand that the female-biased genes are more conserved than the male-biased genes, and this observation is unique. Reports from the developmental expression profile of SOCS genes in the mosquito species *A. culicifacies* and *Anopheles gambiae* suggest a male bias [40,42]. Further, the sexual dimorphism of the JAK-STAT pathway substrate “STAT5” (paralogs STAT5a and STAT5b) has been analyzed in mouse models. STAT5b-deficient mice exhibit sexually dimorphic growth, with downregulation of female-specific proteins and lower gene expression in males [43]. Sex-biased genes and their transcriptional regulation have both been documented in the malaria vector *A. gambiae* using comparative genomics and transcriptomics [42]. TmSOCS transcript expression in immune-related tissues, such as the fat body and gut, was constitutive. It is possible that SOCS proteins exert different biological functions in different tissues. This was also observed for the expression of SOCS6 in the hemocytes, fat body, and Malpighian tubules of *B. mori* larvae [22]. Additionally, higher expression levels of SOCS6 in hemocytes have also been reported in other invertebrate and vertebrate species [20,43,44].

### 3.5. Expression of TmSOCS Genes after Immune Stimulation

In a time-course study conducted over a 2-day period, we examined TmSOCS5, TmSOCS6, and TmSOCS7 mRNA expression in the hemocytes, fat body, and gut tissues of *T. molitor* larvae after infection with *E. coli*, *S. aureus*, or *C. albicans*. The fat body and hemocytes influence the production of inducible immune effectors and phagocytes in insects, and the gut is involved in specific immune reactions. Therefore, these are the preferred tissues for studies of antimicrobial responses and phagocytosis related to innate immunity in insects. TmSOCS5 expression in the fat body was highest at 12 h post–infection with the fungus *C. albicans* (Figure 10A). In the gut tissues (Figure 10B) and hemocytes (Figure 10C), TmSOCS5 expression was induced 6 and 9 h after fungus challenge, respectively. A similar expression profile was noted for TmSOCS6 in the fat body (Figure 10D), gut (Figure 10E), and hemocytes (Figure 10F) of *T. molitor* larvae after microorganism challenge. TmSOCS6 expression was significantly upregulated (*p* < 0.05) in hemocytes at 3 h post-infection. Generally, immune expression at the early stage following infection is useful for mounting an appropriate response to the pro-inflammatory cytokines induced in most tissues; however, in *Bombyx mori*, SOCS6 induction in hemocytes was delayed and was significantly higher at 24 and 48 h post-challenge with the fungus *Beauveria bassiana* and *E. coli*, respectively [22]. In the fat body, however, BmSOCS6 was significantly induced at 24 h post-infection with *E. coli*, which is in agreement with our results. The differential expression profile of SOCS6 mRNA in *T. molitor* and *B. mori* hemocytes after exposure to two different fungal pathogens may explain the role of *C. albicans* as a potent immune elicitor for regulatory genes in insects. Additionally, the relative mRNA expression of EGFR pathway-related genes including *fkhr*, *gsk3*, *ras*, and *erk*, was strongly induced 4 h after injection with recombinant BmSOCS6 protein [22]. Regulation of the EGFR pathway by *Drosophila* SOCS44A (34% identity with human SOCS6 gene) and *B. mori* SOCS6 demonstrate its importance in the control of developmental and pathophysiological processes [22,45]. TmSOCS7 was significantly upregulated in the tested tissues (*p* < 0.05) after challenge with the Gram-negative bacteria *E. coli*, the Gram-positive bacterium *S. aureus*, and the fungus *C. albicans* (Figure 10G–I). As previously observed, *C. albicans* induced a 15-fold increase in TmSOCS7 levels 3 h post-infection. Although studies examining the function of SOCS7 are limited, one study demonstrated that this protein influences STAT3 and STAT5 nuclear translocation [46]. As STAT3 is a transcriptional regulator of IFN-β and interleukin 6, it is understood that SOCS7 participates in the interferon regulatory pathway. The expression of SOCS5, SOCS6, and SOCS7 post-infection with intracellular and extracellular bacterial pathogens has been studied in tongue sole (*Cynoglossus semilaevis*) [47]. CsSOCS7 mRNA was expressed at 6 h when compared with the expression profile of CsSOCS5 and CsSOCS6 mRNA post-infection with bacteria. Further, differential expression of SOCS mRNA was also observed. These results, combined with our observations, suggest that all SOCS genes exhibit enhanced expression post-challenge with microorganisms; however, these expression mechanisms are differentially regulated.

## 4. Conclusions

This study advances our knowledge of insect immunity by identifying and characterizing three type I SOCS gene family members (SOCS5, SOCS6, and SOCS7) in *T. molitor*. We screened *T. molitor* RNA-Seq and Genome-Seq datasets for type I SOCS family members and we examined the evolutionary relationships to other proteins in the type I and type II subfamilies. The observed upregulation of TmSOCS transcripts in the immune-related tissues of *T. molitor* after microbial challenge suggests that they are critical for immune reactions. In the future, we plan to conduct RNA interference experiments to study the involvement of type I SOCS family members in the regulation of key cytokine regulatory pathways. The currently available information regarding insect SOCS genes is very limited, and this study extends the repertoire of possible negative regulators involved in maintaining cellular homeostasis in insects.

## Figures and Tables

**Figure 1 insects-10-00076-f001:**
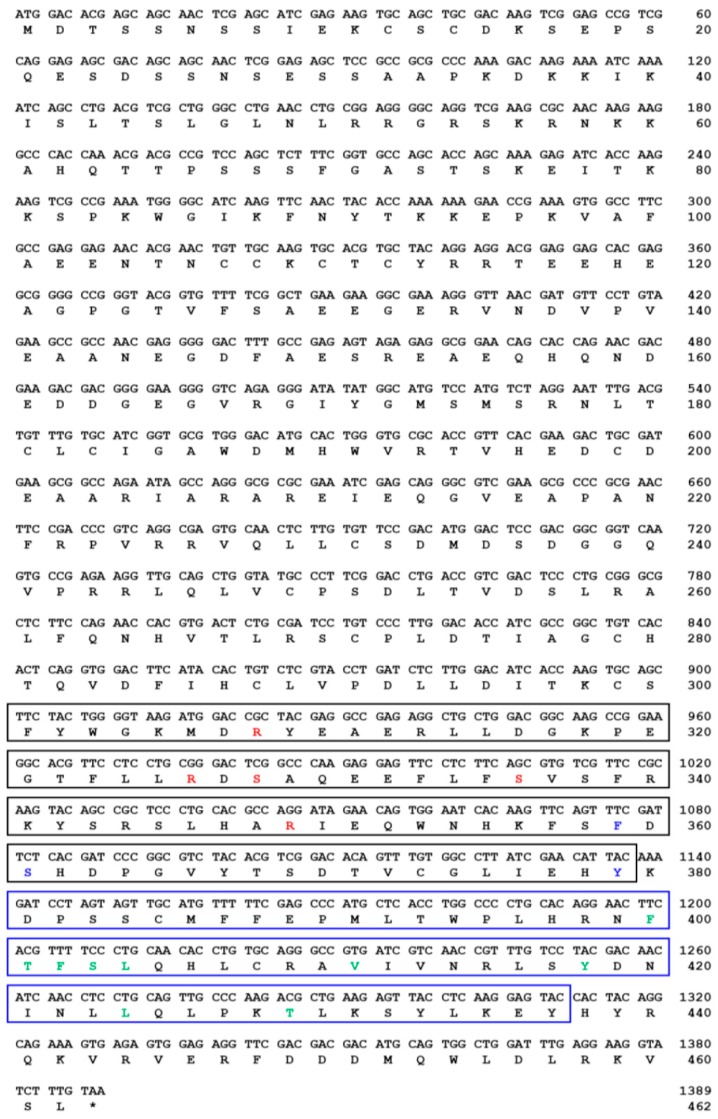
The nucleotide and deduced amino acid sequence of *Tenebrio molitor* SOCS5 (TmSOCS5). The Src Homology 2 (SH2) and suppressor of cytokine signaling (SOCS) box domains are shown in black and blue boxes, respectively. The polypeptide binding sites, including the phosphotyrosine binding pocket, hydrophobic binding pocket, and putative elongin B/C interaction residues, are shown in red, blue, and green text, respectively.

**Figure 2 insects-10-00076-f002:**
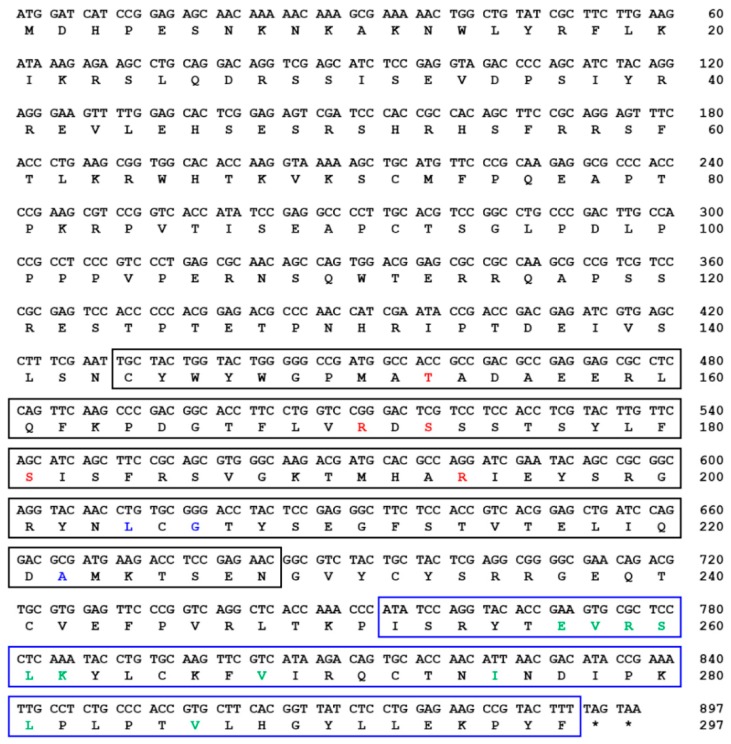
The nucleotide and deduced amino acid sequence of *T. molitor* SOCS6 (TmSOCS6). The SH2 and SOCS box domains are shown in black and blue boxes, respectively. The phosphotyrosine binding pocket, hydrophobic binding pocket, and putative elongin B/C interaction residues are highlighted in red, blue, and green text, respectively.

**Figure 3 insects-10-00076-f003:**
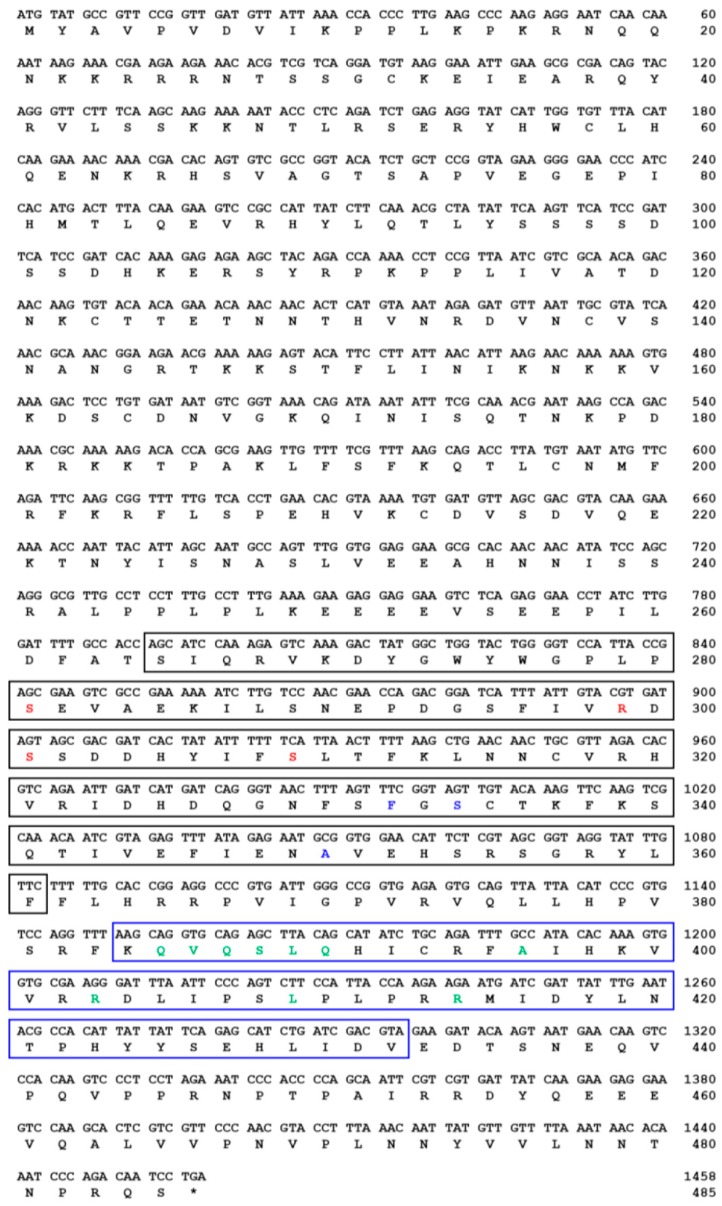
The nucleotide and deduced amino acid sequence of *T. molitor* SOCS7 (TmSOCS7). The conserved SH2 and SOCS box domains are shown in black and blue boxes, respectively. The phosphotyrosine binding pocket, hydrophobic binding pocket, and putative elongin B/C interaction residues are highlighted in red, blue, and green text, respectively.

**Figure 4 insects-10-00076-f004:**
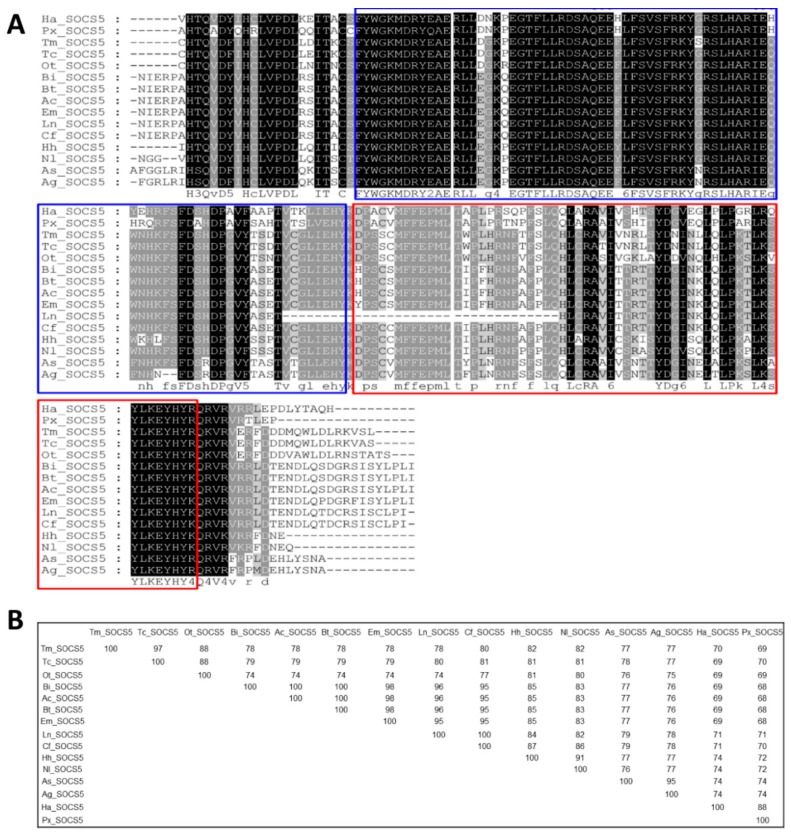
Alignment of TmSOCS5 to other SOCS5 proteins and percent identities. (**A**) Multiple sequence alignment of the SOCS5 C-terminal region of insect orthologs. The SH2 and SOCS box domains are shown in blue and red boxes, respectively. Identical residues in all sequences are shaded black, and well-conserved sequences are shaded grey. Deletions are indicated by dashes; (**B**) Percent identity matrix of TmSOCS5 and representative SOCS5 proteins. The analysis was performed by clustalX2 using representative amino acid sequences from *Tribolium castaneum* (XP_015833443.1), *Plutella xylostella* (XP_011566498.1), *Helicoverpa armigera* (XP_021196679.1), *Onthophagus taurus* (XP_022915891.1), *Bombus impatiens* (XP_012245047.1), *Bombus terrestris* (XP_012163125.1), *Apis cerana* (AEY61566.1), *Eufrisea mexicana* (OAD52276.1), *Lasius niger* (KMQ91264.1), *Camponotus floridanus* (EFN62367.1), *Halyomorpha halys* (XP_014279177.1), *Nilaparvata lugens* (XP_022206249.1), *Anopheles sinensis* (KFB35251.1), and *Anopheles gambiae* (ABV01933.1).

**Figure 5 insects-10-00076-f005:**
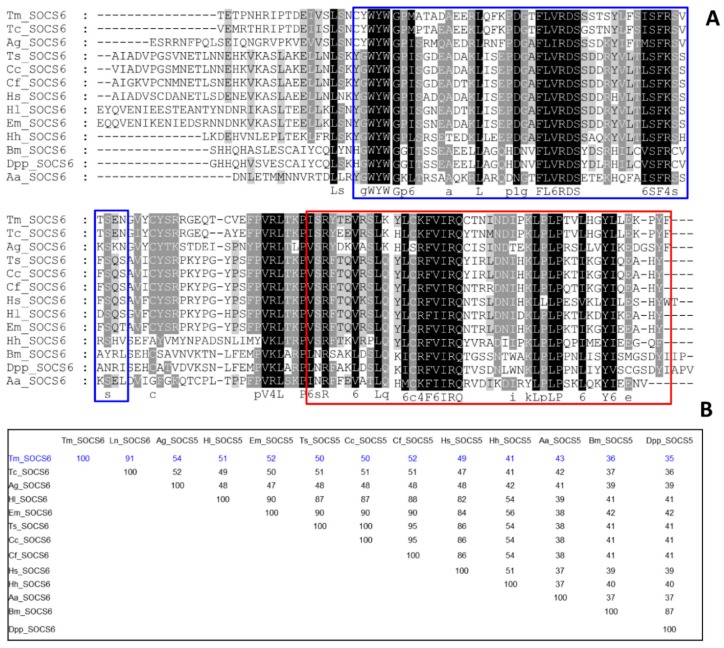
Alignment of TmSOCS6 to other SOCS6 proteins and percent identity: (**A**) Multiple sequence alignment of insect SOCS6 to insect proteins. Only the C-terminal region was conserved. SH2 and SOCS box domains are shown in blue and red boxes, respectively. Identical residues in all sequences are shaded black, and similar sequences are shaded grey. Deletions are indicated by dashes; (**B**) Percent identity matrix of SOCS6 members from representative insect species. Analysis was performed by clustalX2 using representative amino acid sequences from *Tribolium castaneum* (XP_008190646.1), *Anopheles gambiae* (JAB61954.1), *Trachymyrmex septentrionalis* (KYN39913.1), *Ceratitis capitata* (KYN07884.1), *Camponotus floridanus* (EFN74396.1), *Harpegnathos saltator* (EFN81362.1), *Habropoda laboriosa* (KOC63154.1), *Eufrisea mexicana* (OAD52403), *Halyomorpha halys* (XP_014279265.1), *Bombyx mori* (NP_001185652.1), *Danaus plexippus plexippus* (OWR49085.1), *Aedes aegypti* (XP_001660156.3).

**Figure 6 insects-10-00076-f006:**
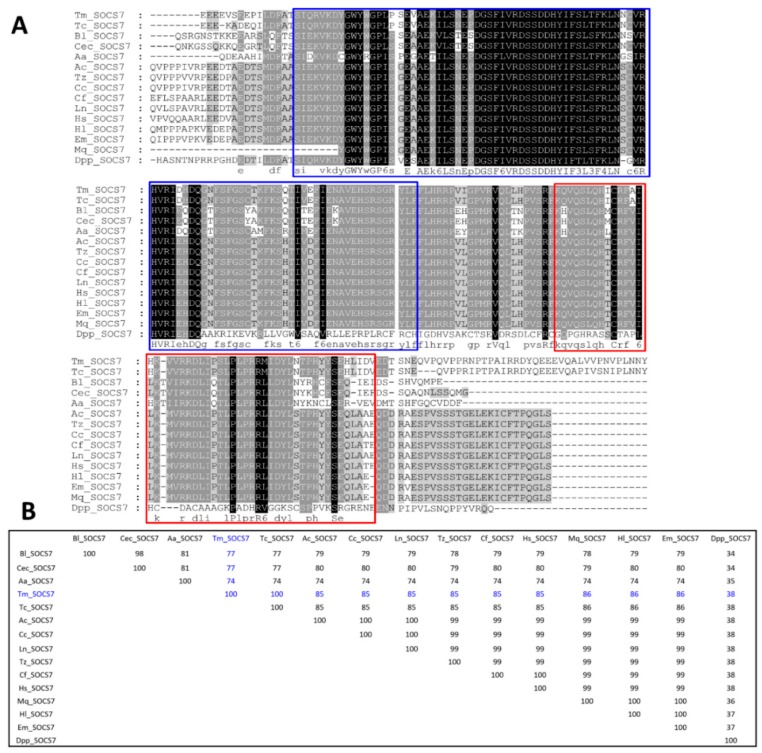
Alignment and identity of TmSOCS7 to other SOCS7 proteins: (**A**) Multiple alignment of the SOCS7 C-terminal sequences of insect orthologs. The SH2 and SOCS box domains are shown in blue and red boxes, respectively. Identical residues in all sequences are shaded black, and similar sequences are shaded grey. Deletions are indicated by dashes; (**B**) Percent identity matrix of SOCS7 proteins using the amino acid sequences from representative insect species. ClustalX2 was used to generate the identity matrix. The sequences used in the analysis were from *Tribolium castaneum* (XP_008190646.1), *Bactrocera latifrons* (JAI49983.1), *Ceratitis capitata* (JAC02138.1), *Aedes aegypti* (XP_021707758.1), *Acromyrmex echinatior* (EGI60822.1), *Trachymyrmex zeteki* (KYQ56054.1), *Cyphomyrmex costatus* (KYM98051.1), *Camponotus floridanus* (EFN69786.1), *Lasius niger* KMQ86622), *Harpegnathos saltator* (EFN83798.1), *Habropoda laboriosa* (KOC61557.1), *Eufrisea mexicana* (OAD58426.1*), Melipona quadrifasciata* (KOX74779.1), and *Danaus plexippus plexippus* (OWR46815.1).

**Figure 7 insects-10-00076-f007:**
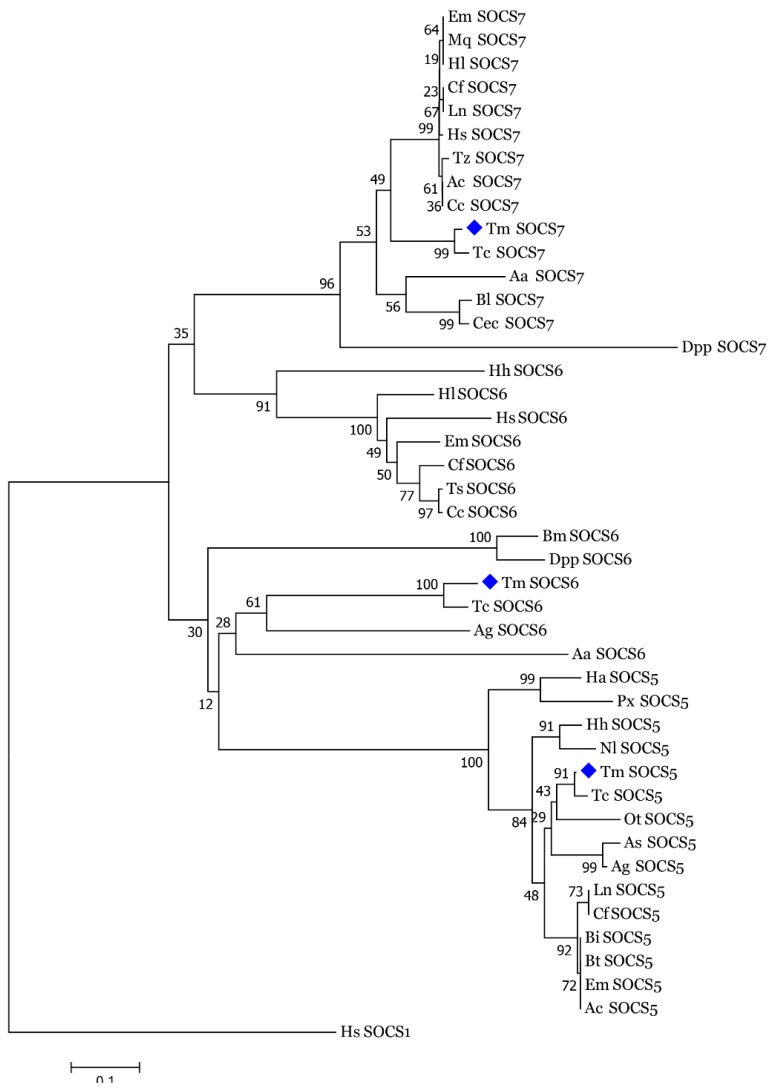
Phylogenetic analysis of TmSOCS5, TmSOCS6, and TmSOCS7 and insect SOCS5, SOCS6, and SOCS7 proteins. The tree was constructed in MEGA7 using the neighbor-joining method. Numbers at the nodes are the bootstrap support of 1000 replicates. *Homo sapiens* SOCS1 (Type II member) was used as an outgroup. 
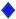
 indicates *T. molitor* SOCS5, SOCS6, and SOCS7 proteins.

**Figure 8 insects-10-00076-f008:**
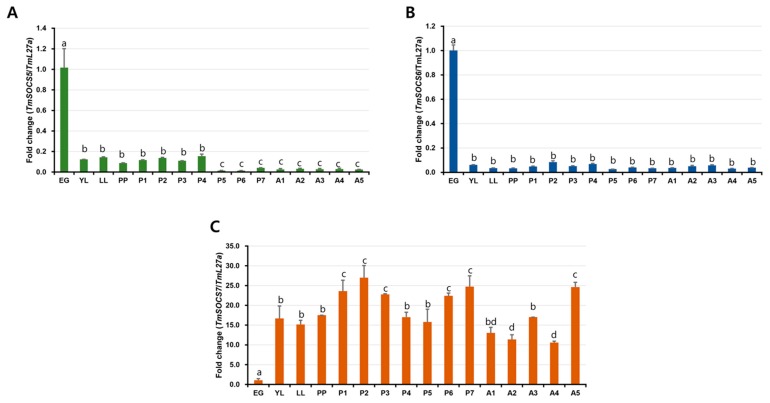
Developmental expression patterns of (**A**) SOCS5, (**B**) SOCS6, and (**C**) SOCS7 transcripts in *T. molitor*. Relative expression of SOCS mRNA in the egg (EG), early larval (YL), late larval (LL), day 1–7 pupal (P1–P7), and day 1–5 adult (A1–A5) stages Transcript levels were analyzed by quantitative real-time PCR. RpL27a (*T. molitor*) was included as an internal control to normalize RNA levels between samples. Results represent mean ± S.E. The experiment was performed three times with similar results. Different letters (a, b, c, d, and bd) above the bars denotes significant differences (*p* < 0.05).

**Figure 9 insects-10-00076-f009:**
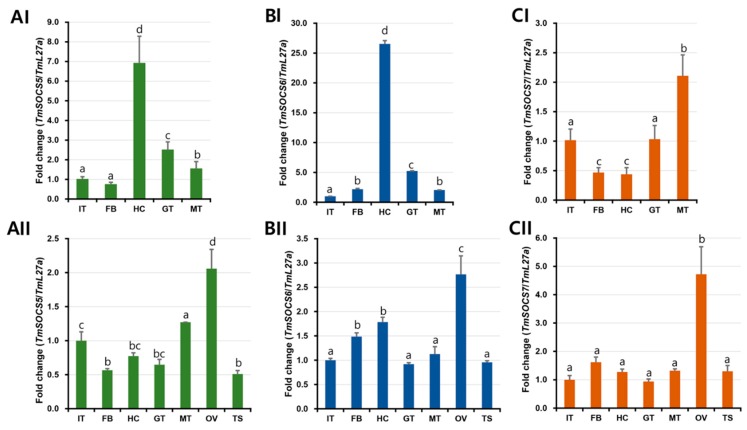
Tissue distribution of TmSOCS transcripts in *T. molitor*: (**A**) Distribution of TmSOCS5 transcripts in larval (AI) and adult tissues (AII); (**B**) distribution of TmSOCS6 transcripts in larval (BI) and adult tissues (BII); (**C**) distribution of TmSOCS7 transcripts in larval (CI) and adult tissues (CII). Tissues are abbreviated as follows: IT, integument; FB, fat body; HC, hemocytes; GT, gut; MT, Malpighian tubules; OV, ovary; TS, testis. RpL27a (*T. molitor*) was included as an internal control to normalize RNA levels among samples. Vertical bars represent mean ± S.E. Different letters (a, b, c, d, and bc above bars) indicate significant differences among groups (*p* < 0.05).

**Figure 10 insects-10-00076-f010:**
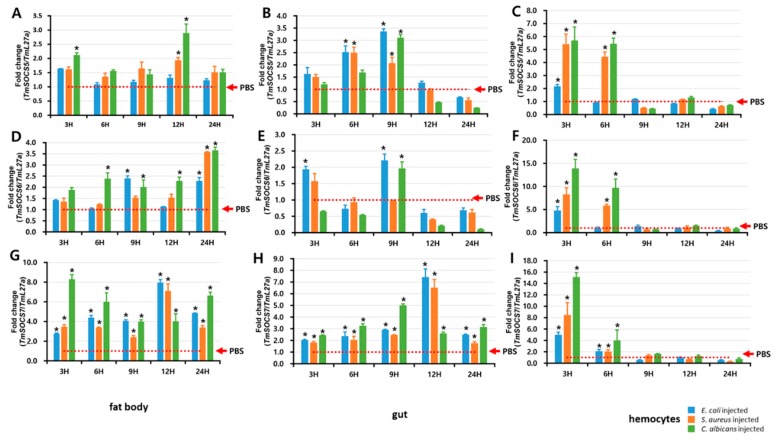
Expression of TmSOCS5, TmSOCS6, and TmSOCS7 mRNA in the fat body (**A**,**D**,**G**), gut (**B**,**E**,**H**), and hemocytes (**C**,**F**,**I**) of *T. molitor* following challenge with *E. coli*, *S. aureus*, and *C. albicans*. Expression was analyzed by real-time PCR using RpL27a as the control for normalization. For each time point, the expression level in the PBS-injected control was set to 1, and this is represented by a dotted line. Values are the mean of three independent measurements ± S.E. (*n* = 3). * *p* < 0.05.

**Table 1 insects-10-00076-t001:** Primers used in this study.

Primer Name	Sequences (5′→3′)
TmSOCS5-cloning_Fw TmSOCS5-cloning_Rv	CCCCCTGAGATGTGATTTCC AACACCGCACATGAAAACAA
TmSOCS5-qPCR_Fw TmSOCS5-qPCR_Rv	CGCGCCCAAAGACAAGAAAATC TTTGGTGGGCCTTCTTGTTG
TmSOCS5-qPCR_Fw2 TmSOCS5-qPCR_Rv2	TCACGTTTTCCCTGCAACAC TGCAGGAGGTTGATGTTGTC
TmSOCS6-cloning_Fw TmSOCS6-cloning_Rv	AGTGTCGGTTGTGCGTGGT GCGCGATTACTAAAAGTACGG
TmSOCS6-qPCR_Fw TmSOCS6-qPCR_Rv	TAAAGAGAAGCCTGCAGGACAG TCCGAGTGCTCCAAAACTTC
TmSOCS7-cloning_Fw TmSOCS7-cloning_Rv	CAGTGTCTCACGATACGCTTTC AGTCTCAGGATTGTCTGGGATT
TmSOCS7-qPCR_Fw TmSOCS7-qPCR_Rv	ATTGAAGCGCGACAGTACAG AAGTCATGTGGATGGGTTCCC
TmL27a_qPCR_Fw TmL27a_qPCR_Rev	TCATCCTGAAGGCAAAGCTCCAGT AGGTTGGTTAGGCAGGCACCTTTA
M13_Fwd M13_Rev	GTAAAACGACGGCCAGT CAGGAAACAGCTATGAC

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
