# Peer review of "Molecular Cloning and Expression Analysis of Three Suppressors of Cytokine Signaling Genes (SOCS5, SOCS6, SOCS7) in the Mealworm Beetle Tenebrio molitor"

_insects, 2019, doi:10.3390/insects10030076_

Round 1
Reviewer 1 Report
Manuscript ID: insects-442498
Title: Molecular cloning and expression characterization of three suppressors of cytokine signaling genes (SOCS5, SOCS6, SOCS7) from the mealworm beetle, Tenebrio molitor
In the present manuscript Patnaik et al. analyze three suppressors of cytokine signaling genes from Tenebrio molitor. After cloning the genes, the authors perform an in silicocharacterization and a phylogenetic analysis of TmSOC5, 5, and 7. Moreover they evaluate the expression of the three genes in the larvae and adult insects. Finally, by challenging the larvae with different pathogens, they conclude that these SOCS are involved in the immune response of the insect.
Overall the study is descriptive. However the rationale is clear and the results are convincing. I think that the authors need to address the following aspects to improve the paper:
1) A language revision is necessary to have a fluent manuscript. Some paragraphs are confused and the meaning of some sentences is unclear.
2) The authors merged Results and Discussion. However I think that the manuscript does not contain a true discussion. The authors just describe results and compare them with previous literature, while they should critically discuss them. For example, why does gene expression decrease suddenly 6 hs after the immune challenge in the hemocytes, while in other tissues it lasts longer? Why are the three genes preferentially expressed in some specific tissues? The authors speculate that SOCS proteins may play different functions in different tissues: is this hypothesis supported by previous studies? Finally, why is SOCS7 highly expressed from YL up to A5, differently from other genes?
3) The last part of the abstract is confused and needs to be rewritten.
4) Lines 39-43. The meaning of this sentence is unclear. Please rephrase.
5) Lines 50-52. Where were SOCS identified first? In vertebrates?
6) Lines 63-66. The expansion of SOCS gene family in relation to the evolution of the rainbow trout SOCS gene family is unclear.
7) Line 85: “in response to microorganisms”
8) Lines 103-104. I did not understand how the authors obtained RNAseq and ETS libraries. This information should be provided.
9) Line 130. Please define better early and late larvae.
10) It is not clear why the authors initially evaluated the expression of SOCS in many tissues while, in immune challenge experiments, only in the fat body, midgut, and hemocytes. Do they exclude an increase in gene expression in the remaining tissues?
11) Lines 188-189. Please mention the related studies.
12) Line 323. I suggest to avoid the term “immune tissues”.
Author Response
Rebuttal letter to the reviewer’s comments on our submitted manuscript (Insects-442498)
Title: Molecular cloning and expression characterization of three suppressors of cytokine signaling genes (SOCS5, SOCS6, SOCS7) from the mealworm beetle, Tenebrio molitor
We, the authors would like to thank you and the reviewers for providing suggested guidance regarding the improvement of the manuscript to journal standards. We have revised few portions of the manuscript as suggested. We have improved the ‘Discussion’ section by adding more references. Subsequently, we have submitted the revised manuscript to English language editing company for a thorough language check, as instructed by the reviewers. We have also improved the clarity of figures as suggested by the reviewers. We were very happy to note that reviewers have appreciated our manuscript in their ratings. As a matter of fact, we were obliged to get such constructive technical advice from the journal ‘Insects’ reviewer panel and feel benefitted.
Please find our response to the comments-
Editor comments:
Reviewer 1:
In the present manuscript Patnaik et al. analyze three suppressors of cytokine signaling genes from Tenebrio molitor. After cloning the genes, the authors perform an in silico characterization and a phylogenetic analysis of TmSOC5, 5, and 7. Moreover they evaluate the expression of the three genes in the larvae and adult insects. Finally, by challenging the larvae with different pathogens, they conclude that these SOCS are involved in the immune response of the insect.
Overall the study is descriptive. However, the rationale is clear and the results are convincing. I think that the authors need to address the following aspects to improve the paper:
Author’s response: We are happy to note the comments of the reviewer. We have further improved the manuscript based on the respected reviewer’s comments.
1) A language revision is necessary to have a fluent manuscript. Some paragraphs are confused and the meaning of some sentences is unclear.
Author’s response: After revising the manuscript, we have sent the manuscript to English language editing services and get the language checked.
2) The authors merged Results and Discussion. However, I think that the manuscript does not contain a true discussion. The authors just describe results and compare them with previous literature, while they should critically discuss them. For example, why does gene expression decrease suddenly 6 hs after the immune challenge in the hemocytes, while in other tissues it lasts longer? Why are the three genes preferentially expressed in some specific tissues? The authors speculate that SOCS proteins may play different functions in different tissues: is this hypothesis supported by previous studies? Finally, why is SOCS7 highly expressed from YL up to A5, differently from other genes?
Author’s response: Many thanks for your suggestion. We have done our best to add discussion points as was referred by the reviewer. Please visit section 3.3 and 3.4 for explanations by referring to the available literature and critical discussions.
3) The last part of the abstract is confused and needs to be rewritten.
Author’s response: We have revised the abstract (especially the last part) as per the suggestions.
4) Lines 39-43. The meaning of this sentence is unclear. Please rephrase
Author’s response: Revised accordingly.
5) Lines 50-52. Where were SOCS identified first? In vertebrates?
Author’s response: In vertebrates. Revised accordingly.
6) Lines 63-66. The expansion of SOCS gene family in relation to the evolution of the rainbow trout SOCS gene family is unclear.
Author’s response: The sentence has been completely revised.
“However, the SOCS gene repertoire is greatly expanded in rainbow trout, with 26 expressed genes in the type I and type II subfamilies. This has been attributed to the expansion of type II SOCS genes from a single CISH/SOCS1-3 precursor via whole genome duplication events.”
7) Line 85: “in response to microorganisms”
Author’s response: Revised as stated.
8) Lines 103-104. I did not understand how the authors obtained RNAseq and ETS libraries. This information should be provided.
Author’s response: T. molitor RNAseq and EST libraries are not published. But ‘Chonnam National University researchers’ have the libraries and hence they screened the SOCS genes using T. castaneum SOCS genes as queries.
9) Line 130. Please define better early and late larvae.
Author’s response: We have defined early and late larvae in the section 2.5.
10) It is not clear why the authors initially evaluated the expression of SOCS in many tissues while, in immune challenge experiments, only in the fat body, midgut, and hemocytes. Do they exclude an increase in gene expression in the remaining tissues?
Author’s response: Of course, a tissue-specific expression was suggested in Figure 8, 9, and 10. As our goal was to study SOCS in the perspective of innate immunity, we were interested to understand the immune induction in hemocytes, gut, and fat body of T. molitor larvae. We have succinctly discussed the same in section 3.3 and 3.4.
11) Lines 188-189. Please mention the related studies.
Author’s response: We have added the related studies as references [30,31].
12) Line 323. I suggest to avoid the term “immune tissues”.
Author’s response: Revised the same as advised.
We sincerely hope that we have addressed the suggestions and comments as best we can. If there are further suggestions, we would like to work on the same.
Thanking you
Sincerely,
In Seok Bang, PhD. (Corresponding Author)

Reviewer 2 Report
Dear Author/s.
Characterization of three SOCS genes (SOCS5, SOCS6 and SOCS7) in silico and in vivo infection of (Gram +ve, -ve bacteria and fungus) itself is a huge amount of work and eventually data presentation again a big deal. Hence, I am suggesting some of the correction to make the report concise and meaningful.
Major issues: Justify with the answers and/or make the necessary changes in the manuscript.
Abstract: In page 1 line no. 23 “While the expression of TmSOCS5 and TmSOCS6 was low in larval, pupal and adult stages of the insect, TmSOCS7 showed higher expression” Whether expression of SOCS5, SOCS6 and SOCS7 were compared among themselves?
Another statement lin no 26 “The microbes expressed the three TmSOCS genes to varying degrees.” Whether microbial SOCS gene expression was analyzed? Make the necessary changes, which should also rational to the results.
Introduction: Reference no 6-11 related to the human/vertebrate SOCS, used in second paragraph of introduction for describing the different mechanism of SOCS homologs in negative regulation of JAK-STAT pathway; author/s has nowhere mentioned about the host species.
Material and Methods: Section 2.5. (Developmental and Tissue-specific expression of TmSOCS transcripts) Extension time, Fluorescence readings temperature and final extension is missing in the real-time qPCR cycle condition.
Section 2.6. (Immune challenge studies): Author/s did not mention, how much volume of PBS suspended microbes, did they injected to each larva from the 106 cells/µl of E. coli and S. aureus and 5 x 104 cells/µl of C. albicans. Correspondingly, it is also necessary to mention that, which body part of the larva was used for administration of immune elicitor/s and justify why only fat bodies, gut and hemocytes tissues were collected for observing the differential change in expression at various time-interval (for kinetic study).
Results: Section 3.1 (Identification of TmSOCS homologs and molecular characterization): Author/s has provided the six independent figures for this result section, which is too much. It is a valuable suggestion to author/s that submit the full SOCS genes (SOCS5, SOCS6 and SOCS7) sequences (nucleotide and protein) to GenBank at NCBI and provide the accession number in this section. Moreover, in place of that, for describing the SOCS genes (SOCS5, SOCS6 and SOCS7), prepare the genomic and proteomic architectures (through graphical presentation) and assemble all of them in one figure with their exonic, intonic and N-terminal region, SH2 and SOCS box domains respectively. As far as the some conserved amino acid residue (polypeptide binding sites, namely the phosphotyrosine binding pocket, hydrophobic binding pocket and putative elongin B/C interaction) is concerned, they can make it visible over the multiple sequence alignment (MSA) of SOCS genes (SOCS5, SOCS6 and SOCS7) in figure 4, 5 and 6 respectively. This will be more accurate and precise.
Insect species chosen for the MSA from Genbank; whether all of them have already characterized/named as SOCS5, SOCS6 and SOCS7 in NCBI database. How did the author recognize the identity of SOCS gene (5, 6 and 7) in different insect species?
Why did author has chosen only the conserved SH2 and SOCS-box domain not the N-terminal region, which also has the conserved domain recognized as NTCR (Chandrashekaran, et al., 2015 Structure and functional characterization of the conserved JAK interaction region in the intrinsically disordered N‑terminus of SOCS5. Biochemistry 54:4672-4682 and Feng et al., 2012 The N-terminal domains of SOCS proteins: a conserved region in the disordered N-termini of SOCS4 and 5. Proteins 80:946-957.)
Prepare the MSA figures (fig no 4A and 6A) more clearly for SOCS5 and SOCS7; as has been done for SOCS6. All of them should have uniform appearance.
Result Section 3.3 (The developmental and tissue distribution of TmSOCS gene expression) and 3.4 (Expression of TmSOCS genes after immune stimulation): Author/s has mentioned, that “the one way analysis of variance (ANOVA) and Tukey’s multiple range tests” was performed but did not revealed the platform or software name which they used. Whether all of the real time PCR expression analysis (graphs and statistics) was done using the same platform? Figure 8A, 9A and 10A; why did the author/s has chosen so many different stages (P1-P7 and A1-A5) for each day of developmental cycle and presenting all of them; even when there is no significant changes were found between them?
Similarly, Tissue specific expression of SOCS5, SOCS6 and SOCS7 in testis and ovary showed a significant difference (Figure 8C, 9C and 10C) is there any role in sexual dimorphisms of male and female played by these gene not been mentioned and discussed anywhere. Refer to (Magnusson et al., 2011. Transcription regulation of sex-biased genes during ontogeny in the malaria vector Anopheles gambiae. PLoS One 6, e21572.) Correspondingly, In the expression analysis of developmental stages of Tenebrio A1-A5 or P1-P7 whether the collected samples were male adult and pupae or female? Need to correlate the results of 8C, 9C and 10C with 8A, 9A and 10A respectively. Representation of statistical values (a, b, c, d on bar graphs) on these results are not convincing and even not well described in the figure legend.
Result section 3.4 (Expression of TmSOCS genes after immune stimulation): Figure 11, 12 and 13, which showed the time kinetic study of different types of immune challenges (Gram +ve, -ve bacteria and fungus) showed significant differences within 12 hours. Author/s did not describe completely the findings of immune elicitors at various time points for which they collected the samples. It is meaningless to put all of the experimental results in the figures, until unless you are not describing each of them in the text of a research article (like time-points and in various collected tissues).
Hence, it is advisable that analyze the results one more time and describe them consequently. Assemble all of these figures (Figure 11, 12 and 13) in to one panel accordingly for all three SOCS genes (SOCS5, SOCS6 and SOCS7). However, figure legends of these figures almost same, hence no need to repeat them again and again, change it accordingly.
Consolidated and amalgamated seven figures will make the manuscript more reader friendly and technically sound. Expecting the required changes would have incorporated in the manuscript to make the manuscript and figures crispy and rational.
With Best Wishes.
Author Response
Rebuttal letter to the reviewer’s comments on our submitted manuscript (Insects-442498)
Title: Molecular cloning and expression characterization of three suppressors of cytokine signaling genes (SOCS5, SOCS6, SOCS7) from the mealworm beetle, Tenebrio molitor
We, the authors would like to thank you and the reviewers for providing suggested guidance regarding the improvement of the manuscript to journal standards. We have revised few portions of the manuscript as suggested. We have improved the ‘Discussion’ section by adding more references. Subsequently, we have submitted the revised manuscript to English language editing company for a thorough language check, as instructed by the reviewers. We have also improved the clarity of figures as suggested by the reviewers. We were very happy to note that reviewers have appreciated our manuscript in their ratings. As a matter of fact, we were obliged to get such constructive technical advice from the journal ‘Insects’ reviewer panel and feel benefitted.
Please find our response to the comments-
Editor comments:
Reviewer 2:
Dear Author/s.
Characterization of three SOCS genes (SOCS5, SOCS6 and SOCS7) in silico and in vivo infection of (Gram +ve, -ve bacteria and fungus) itself is a huge amount of work and eventually data presentation again a big deal. Hence, I am suggesting some of the correction to make the report concise and meaningful.
Author’s response: Many thanks for your praise. We have tried to comply to your suggestions in this revision. As suggested by you, we have made thorough language corrections.
Major issues: Justify with the answers and/or make the necessary changes in the manuscript.
Abstract: In page 1-line no. 23 “While the expression of TmSOCS5 and TmSOCS6 was low in larval, pupal and adult stages of the insect, TmSOCS7 showed higher expression” Whether expression of SOCS5, SOCS6 and SOCS7 were compared among themselves?
Author’s response: We have revised the abstract.
Another statement line no 26 “The microbes expressed the three TmSOCS genes to varying degrees.” Whether microbial SOCS gene expression was analyzed? Make the necessary changes, which should also rational to the results.
Author’s response: This was a mistake on our part. We have revised the sentence accordingly.
Introduction: Reference no 6-11 related to the human/vertebrate SOCS, used in second paragraph of introduction for describing the different mechanism of SOCS homologs in negative regulation of JAK-STAT pathway; author/s has nowhere mentioned about the host species.
Author’s response: We have made the necessary corrections in the Introduction section.
Material and Methods: Section 2.5. (Developmental and Tissue-specific expression of TmSOCS transcripts) Extension time, Fluorescence readings temperature and final extension is missing in the real-time qPCR cycle condition.
Author’s response: We have added the necessary information in section 2.5 as suggested.
Section 2.6. (Immune challenge studies): Author/s did not mention, how much volume of PBS suspended microbes, did they injected to each larva from the 106 cells/µl of E. coli and S. aureus and 5 x 104 cells/µl of C. albicans. Correspondingly, it is also necessary to mention that, which body part of the larva was used for administration of immune elicitor/s and justify why only fat bodies, gut and hemocytes tissues were collected for observing the differential change in expression at various time-interval (for kinetic study).
Author’s response: 1µl suspension containing microorganisms was injected (intra-abdominal). Similar volume of PBS was also injected. We have revised accordingly in section 2.6.
Results: Section 3.1 (Identification of TmSOCS homologs and molecular characterization): Author/s has provided the six independent figures for this result section, which is too much. It is a valuable suggestion to author/s that submit the full SOCS genes (SOCS5, SOCS6 and SOCS7) sequences (nucleotide and protein) to GenBank at NCBI and provide the accession number in this section. Moreover, in place of that, for describing the SOCS genes (SOCS5, SOCS6 and SOCS7), prepare the genomic and proteomic architectures (through graphical presentation) and assemble all of them in one figure with their exonic, intonic and N-terminal region, SH2 and SOCS box domains respectively. As far as the some conserved amino acid residue (polypeptide binding sites, namely the phosphotyrosine binding pocket, hydrophobic binding pocket and putative elongin B/C interaction) is concerned, they can make it visible over the multiple sequence alignment (MSA) of SOCS genes (SOCS5, SOCS6 and SOCS7) in figure 4, 5 and 6 respectively. This will be more accurate and precise.
Author’s response: The sequences have been registered with GenBank. The accession numbers are MK292064, MK292065, and MK292066 for TmSOCS5, TmSOCS6, and TmSOCS7, respectively. We haven’t made a fosmid library so that we could provide the genomic architecture (including the exonic and intronic regions) of T. molitor SOCS genes. In this case, we have cloned only the ORF sequence and hence, have characterized the gene and the translated protein sequence. Hence, the suggestions have been noted and would be favourably considered in the future.
Insect species chosen for the MSA from Genbank; whether all of them have already characterized/named as SOCS5, SOCS6 and SOCS7 in NCBI database. How did the author recognize the identity of SOCS gene (5, 6 and 7) in different insect species?
Author’s response: The identities of SOCS genes have been revealed in the GenBank and therefore we have placed the characterized names in MSA or phylogenetic analysis.
Why did author has chosen only the conserved SH2 and SOCS-box domain not the N-terminal region, which also has the conserved domain recognized as NTCR (Chandrashekaran, et al., 2015 Structure and functional characterization of the conserved JAK interaction region in the intrinsically disordered N‑terminus of SOCS5. Biochemistry 54:4672-4682 and Feng et al., 2012 The N-terminal domains of SOCS proteins: a conserved region in the disordered N-termini of SOCS4 and 5. Proteins 80:946-957.)
Author’s response: Actually, we have tried to consider the N-terminal region, but the analysis at the sequence level was not true. This is because of sequence divergence at N-terminal regions. However, as suggested, we have discussed the same in the revised manuscript and added the suggested references.
Prepare the MSA figures (fig no 4A and 6A) more clearly for SOCS5 and SOCS7; as has been done for SOCS6. All of them should have uniform appearance.
Author’s response: We have improved the clarity of Figure 4A and 6A.
Result Section 3.3 (The developmental and tissue distribution of TmSOCS gene expression) and 3.4 (Expression of TmSOCS genes after immune stimulation): Author/s has mentioned, that “the one-way analysis of variance (ANOVA) and Tukey’s multiple range tests” was performed but did not revealed the platform or software name which they used. Whether all of the real time PCR expression analysis (graphs and statistics) was done using the same platform? Figure 8A, 9A and 10A; why did the author/s has chosen so many different stages (P1-P7 and A1-A5) for each day of developmental cycle and presenting all of them; even when there is no significant changes were found between them?
Author’s response: We have mentioned the platform or software in the revised version for all the statistical analysis. Basically, we have been rearing the T. molitor larvae and hence could find all the developmental stages for the experiments. We have discussed the same in the ‘Results and Discussion’ section.
Similarly, Tissue specific expression of SOCS5, SOCS6 and SOCS7 in testis and ovary showed a significant difference (Figure 8C, 9C and 10C) is there any role in sexual dimorphisms of male and female played by these genes not been mentioned and discussed anywhere. Refer to (Magnusson et al., 2011. Transcription regulation of sex-biased genes during ontogeny in the malaria vector Anopheles gambiae. PLoS One 6, e21572.) Correspondingly, In the expression analysis of developmental stages of Tenebrio A1-A5 or P1-P7 whether the collected samples were male adult and pupae or female? Need to correlate the results of 8C, 9C and 10C with 8A, 9A and 10A respectively. Representation of statistical values (a, b, c, d on bar graphs) on these results are not convincing and even not well described in the figure legend.
Author’s response: We have discussed sexual dimorphism under section 3.3 as suggested and added the necessary references. At the developmental stage, the adults/pupae were not sexed. We have revised the explanation of statistical values in the figure legend.
Result section 3.4 (Expression of TmSOCS genes after immune stimulation): Figure 11, 12 and 13, which showed the time kinetic study of different types of immune challenges (Gram +ve, -ve bacteria and fungus) showed significant differences within 12 hours. Author/s did not describe completely the findings of immune elicitors at various time points for which they collected the samples. It is meaningless to put all of the experimental results in the figures, until unless you are not describing each of them in the text of a research article (like time-points and in various collected tissues).
Hence, it is advisable that analyze the results one more time and describe them consequently. Assemble all of these figures (Figure 11, 12 and 13) in to one panel accordingly for all three SOCS genes (SOCS5, SOCS6 and SOCS7). However, figure legends of these figures almost same, hence no need to repeat them again and again, change it accordingly.
Author’s response: As suggested, we have extended the discussion section as per the valued suggestions of the reviewers. Further, as suggested, we have made a single panel of figure 11, by merging Figure 11, 12, and 13.
Consolidated and amalgamated seven figures will make the manuscript more reader friendly and technically sound. Expecting the required changes would have incorporated in the manuscript to make the manuscript and figures crispy and rational.
Author’s response: As explained, we have merged Figure 11, 12, and 13 to one figure. But, we have kept the sequence figures as these are novel sequences with GenBank accession numbers.
We sincerely hope that we have addressed the suggestions and comments as best we can. If there are further suggestions, we would like to work on the same.
Thanking you
Sincerely,
In Seok Bang, PhD. (Corresponding Author)

Round 2
Reviewer 1 Report
Manuscript ID: insects-442498
Title: Molecular cloning and expression characterization of three suppressors of cytokine signaling genes (SOCS5, SOCS6, SOCS7) from the mealworm beetle, Tenebrio molitor
This is the revised version of a manuscript by Patnaik et al. in which the authors analyze three suppressors of cytokine signaling genes from Tenebrio molitor.
The authors addressed all the questions raised by the referee and significantly improved the manuscript.
I have only three minor suggestions:
1) Line 76: “immune-related”
2) Line 146: remove the hyphen
3) Line 360. “after microorganism challenge”
Author Response
Rebuttal letter to the reviewer’s comments on our submitted manuscript (Insects-442498R2)
Title: Molecular cloning and expression characterization of three suppressors of cytokine signaling genes (SOCS5, SOCS6, SOCS7) from the mealworm beetle, Tenebrio molitor
We, the authors thank the reviewers for providing suggested guidance regarding the improvement of the manuscript to journal standards. This is the second round of revisions and we have done our best to justify the points raised by the reviewers. We have further improved the ‘Results & Discussion’ section. We have consolidated the figures as suggested by reviewer 2. Finally, the manuscript has been edited and proof-read by an English language editing company. We were happy to note that reviewers have appreciated our manuscript in their ratings. As a matter of fact, we were obliged to get such constructive technical advice from the journal ‘Insects’ reviewer panel and feel benefitted.
Please find our response to the comments-

Reviewer 2 Report
Dear author/s,
The revised version of manuscript (manuscript ID insects-442498 revised1) “Molecular cloning and expression analysis of three suppressors of cytokine signaling genes (SOCS5, SOCS6, SOCS7) in the mealworm beetle Tenebrio molitor” after modification still have some major discrepancies, try to figure them out or justify with the appropriate answers.
1. Method Section 2.6. (Immune challenge studies) Line no. 152-154 “1-µl suspension containing either 106 cells of E. coli or S. aureus or 5 x 104 cells of C. albicans to each larva” is the same dose used for adults stages as well, not been mention in the method. If this is the dose given it to the tiny larva then what is the mortality rate, similarly, cite the reference for the same.
2. Results, section 3.2- I am pointing out one more time, prepare the multiple sequence alignment (MSA) figures (figure no 4A and 6A) more clearly for SOCS5 and SOCS7; as has been done for SOCS6. All of them should have uniform appearance.
3. Results, section 3.3-author has given the explanation that “we have been rearing the T. molitor larvae and hence could find all the developmental stages for the experiments. This is neither looking rational explanation, nor being appropriately justify with the provided results and discussion. I will again give emphasis for the developmental stages (PP to P7 i.e. eight stages and A1 to A5, five stages neither being mentioned as male nor female) for SOCS5, SOCS6 genes, do the necessary corrections and organize them accordingly, because it is not showing any major changes between the pupae (P1-P7) and adults (A1-A5) stage of T. molitor.
4. For TmSOCS7 (line no. 300-301) author has stated in one line “In contrast, TmSOCS7 expression levels in the larval, pupal, and adult stages were much higher than during the egg stage (Figure 10)”. This need more explanation and mention that figure 10A. For the same reason, consider this for other places as well (in figure 8A and 9A).
5. N=3 is the number of individuals tissues (such as gut, Malphigian tubules, fat body, integument and hemocytes) were collected and pooled (n = 3) from larvae and adults for expression analysis or experimental mean of 3 independent experiment? Please make it clear somewhere in the methods.
6. Line no. 294-296 is not clear.
7. Line no. 334-336 – “The tissue expression analysis of T. molitor larvae showed that the mRNA levels of TmSOCS5 and TmSOCS6 were higher in the hemocytes than in the other tissues, whereas TmSOCS7 levels were elevated in Malpighian tubules”. There are three graphs have provided in three different figures (8B, 9B, 10B) related to this line. First relate with them (provide the figure number) then it is crucial to discuss little bit about the findings.
8. Line no. 338 “No previous report has studied the differential SOCS gene expression in males and females” is not correct, kindly refer to Dhawan et al. 2015 Molecular characterization of SOCS gene and its expression analysis on Plasmodium berghei infection in Anopheles culicifacies. In figure 4 and 5 they have emphasize the findings of involvement of SOCS gene in sexual dimorphism.
9. Line no. 345 missing the preposition “in” in the line.
10. In figure numbers 8, 9 and 10, figure legends states that “Different lowercase letters indicate significant differences among groups (p<0.05)” is not clear (Lines no. 315, 323-324 and 332). Representation of statistical values on bar graphs still not convincing, describe somewhere in the figure legends about the denotation of a, b, c, d, bc and bd.
11. Overall, It is my suggestion that, it would be more convincing and better, if author would split the result section 3.3, in two parts (as 3.3 and 3.4) and 3 figures (8, 9 and 10) in to two panels; viz. Expression of SOCS in T. molitor developmental stages and tissue specific expression of SOCS genes. Then, it would be easier for them to present and discuss the results, statistical analysis, to make the comparison as well as description of statistics in each figure legends accordingly.
12. Discussion is the very crucial part for any research paper. It would be very awkward for any reviewer to provide the discussion material every time as stated in line no. 338 (No previous report has studied the differential SOCS gene expression in males and females), which is not correct. Only extended results does not provide the exact discussion, do some more rigorous literature survey.
The authors claim having the manuscript reviewed by English language editing company for a thorough language check. Unfortunately, the company did a bad job. There are still sentences that does not make sense, and have missing words. I will highly advise to the authors to get the manuscript professionally edited and proofread one more time. A revision, with a clearly understandable manuscript addressing the issues pointed above could certainly improve the quality of this piece of work.
I am again highlighting that consolidated figures and balanced discussion will make the manuscript more reader friendly and technically sound. Peer-review process of the journal is for betterment for articles and an effort to deliver the best work out of it before circulating any reports to scientific community; hence, no need to write the rebuttal just explanation of the question is sufficient.
Expecting the author/s would have seriously consider the suggestions and do the required and substantial changes in the manuscript. All the best.
Author Response

(The authors gave the same response as above.)

Round 3
Reviewer 2 Report
I appreciate that author has accepted and followed the suggestions. Manuscript has amended significantly by the authors. Final form of paper is appears upgraded now.
I will again advice to the authors that recheck the paper (read thoroughly and carefully) one more time at your end and minimize the mistakes such as mentioned below.
· Line no. 81- The type I SOCS genes; however have not been previously reported in insects. It should be coleopteran insect or modify accordingly.
· Reference no. 29 (Jo et al. 2017) for the microorganism preparation cite that in the methodology section as well.
Validate all the minor mistakes attentively throughout the manuscript at your level.
I wish author/s will definitely elucidate/expand their research in same direction for other genes (like PIAS etc.) of JAK-STAT or other immune signaling pathway of insects in future as well.
All the best for your upcoming research.